# Kitchen Diet vs. Industrial Diets—Impact on Intestinal Barrier Parameters among Stroke Patients

**DOI:** 10.3390/ijerph19106168

**Published:** 2022-05-19

**Authors:** Maja Czerwińska-Rogowska, Karolina Skonieczna-Żydecka, Krzysztof Kaseja, Karolina Jakubczyk, Joanna Palma, Marta Bott-Olejnik, Sławomir Brzozowski, Ewa Stachowska

**Affiliations:** 1Department of Human Nutrition and Metabolomics, Pomeranian Medical University, 71-460 Szczecin, Poland; majaczerwinska89@gmail.com (M.C.-R.); jakubczyk.kar@gmail.com (K.J.); 2Department od Biochemical Sciences, Pomeranian Medical University, 71-460 Szczecin, Poland; karolina.skonieczna.zydecka@pum.edu.pl (K.S.-Ż.); joanna.palma@pum.edu.pl (J.P.); 3Department of General and Transplant Surgery, Pomeranian Medical University, 71-460 Szczecin, Poland; krzysztof.kaseja@pum.edu.pl; 4Neurology Department Regional Specialist Hospital in Gryfice, 72-300 Gryfice, Poland; sekretariat@lesnapolana24.pl (M.B.-O.); sekretariat@medicam.pl (S.B.)

**Keywords:** enteral nutrition, mixed kitchen diet, ischemic stroke, malnutrition

## Abstract

Background and aims: Strokes are the second highest cause of death in the world and the most common cause of permanent disability in adults. Intestinal barrier permeability thus contributes to diminished homeostasis within the body, which further affects the healing process and convalescence. Each stroke patient should be administered with ingredients that support the intestinal barrier (e.g., protein and fiber). The aim of this study was to compare the effect of various types of diet (enteral with or without fiber vs. a mixed kitchen diet) on the metabolic activity of intestinal microbiota, namely short chain fatty acids, and gut barrier integrity parameters (zonulin and calprotectin. Methods: Patients (*n* = 59), after suffering an ischemic stroke, were randomly allocated to three groups receiving: the kitchen diet (*n* = 32; 1.2 g fiber in 100 mL); Nutrison Energy^®^ (*n* = 14; 0.02 g fiber in 100 mL); and Nutrison Diason Energy HP^®^ (*n* = 13; 1.8 g fiber in 100 mL). The patients underwent anthropometric measurements and blood samples (for prealbumin measurements), and stool samples (for zonulin and calprotectin determinations) were taken twice, on admission and a week later. Results: Industrial diets enriched with fiber maintained nutritional status and had a beneficial effect on intestinal barrier permeability parameters. Patients fed with kitchen diets demonstrated a decreased number of lymphocytes, hemoglobin, erythrocytes, and increased serum concentration of C-reactive protein, but improved gut barrier markers. Proton pump inhibitors were shown to increase the inflammatory process in gut. Conclusions: Stroke patients should be administered with industrial diets enriched with fiber to improve gut barrier integrity and nutritional parameters.

## 1. Introduction

The intestinal barrier consists of a single layer of epithelial cells (mainly enterocytes) connected by tight junctions (TJ), which seal the intercellular space [1] Absorption in the intestinal epithelium is highly selective. Such selective absorption is an unquestionable advantage of the intestinal barrier and it is closely linked to its complex structure. Dysfunction of the intestinal barrier, or intestinal hyperpermeability, also known as the leaky gut syndrome, involves the loosening of TJ secondary to gut microbiota alterations [2]. A biomarker of the intestinal barrier integrity is protein-zonulin, which reversibly regulates intestinal permeability by modulating intercellular TJ. In contrast, fecal calprotectin is a useful marker of intestinal inflammation. This protein is secreted extracellularly from neutrophils related to calprotectin [3,4]. There is a body of evidence that gut integrity in critically ill patients, including those who experienced a stroke, is impaired, which might deteriorate the healing process to a large extent [5]. Stroke patients who are unconscious or have significant difficulties in swallowing should be subjected to enteral nutrition. Enteral nutrition diets might be of a kitchen variety (also called a blended diet or a home diet) or the industrial type [6]. To date, there have been few studies evaluating the effects of enteral nutrition on intestinal barrier function.

Fiber in diet is crucial for gut health. The main fiber fermentation products from are the short chain fatty acids (SCFAs), acetate butyrate, and propionate [7]. The acids serve as sources of energy for colon cells. Positive correlation was found between the consumption of fiber and the fecal levels of SCFAs [8]. In contrast, branched short-chain fatty acids (BCFA), predominantly isovaleric and isobutyric, are markers of protein fermentation. More studies are needed to understand the role of BCFA in human health; however, a negative correlation between the consumption of insoluble fiber and the fecal levels of BCFA was demonstrated, which might indicate that protein degradation is of significant importance to colon cells [9]. Zonulin is a popular serological marker to assess the integrity of the intestinal mucosal barrier. The dysregulation of the zonulin pathway has been associated with irritable bowel syndrome and inflammatory bowel disease [10].

The effects of enteral nutrition following a stroke and its effect on the intestinal barrier permeability have not been fully explored. We aimed to compare the effects of a various types of diet on intestinal SCFAs concentration, gut barrier permeability, and inflammatory parameters.

## 2. Methods

### 2.1. Patients

The study was approved by the Bioethics Committee of the Pomeranian Medical University in Szczecin (consent no. KB-0012/84/16, dated 27 June 2016). All procedures were carried out in compliance with the Declaration of Helsinki. The study group was recruited from Caucasian patients who had suffered an ischemic stroke. The stroke was diagnosed by a medical doctor who performed and evaluated necessary examinations to confirm the diagnosis. Patients were included in the study group if they were anticipated to require an enteral tube feeding for at least 7 days (Figure 1). The following exclusion criteria were applied: long-term enteral nutrition, dysphagia due to non-neurological reasons, comorbid chronic intestinal disease, comorbid cancer, extreme undernourishment, and autoimmune disease.

Nutritional status was evaluated through anthropometric measurements: height, calf and arm circumference (anthropometric tape measure), and subscapular skinfold thickness (Saehan metal skinfold calliper—Baseline, Korea). As all patients were in a lying position, their parameters were estimated based on other measurements. Stature was predicted relative to the distance from the top to bottom of medial tibia, using the knee height equation by Chumlea et al. [11] for Caucasian populations [11]. In turn, body weight was predicted by measuring calf and arm circumferences, knee length, and subscapular skinfold thickness (according to the equation by Chumlea et al.) [12]. The basal metabolic rate and body composition were assessed by bioelectric impedance using a BIA Akern device (Cosmed, Italy).

Patients’ blood biochemical parameters and anthropometric measurements (Table 1) did not differ significantly by diet type.

### 2.2. Types of Nutritional Support

Patients participating in the study were divided into three groups. The first group (*n* = 32) received a blenderized kitchen diet designed by a dietician (using the Esculap software application, in conjunction with the database of the National Food and Nutrition Institute, Warsaw, Poland). All patients on the kitchen diet were given the same formula, with the volume depending on their individual calorie needs. On average, the daily intake amounted to about 2000 mL (four meals). The kitchen diet was based on natural, easily digestible, whole food ingredients, with identical composition and calorie content every day.

Other patients participating in the study received one of two commercial formulas: Nutrison Energy^®^ (Nutricia- without fiber; *n* = 14) and Nutrison Diason Energy HP^®^ (Nutricia- with fiber; *n* = 13). Both industrial diets had a similar energy content. The volume of formula depended on individual patient’s needs. On average, it was 1000–1500 mL per day (Figure 1).

### 2.3. Blood Biochemistry

Biochemical measurements were made for cholesterol, triglycerides, and complete blood count. As an additional test, serum prealbumin concentration was determined. The latter was carried out by ELISA (Immundiagnostik, Germany), according to the manufacturer’s instructions. The measurements were taken on admission and a week after the enteral nutrition started.

### 2.4. Isolation and Measurement of SCFAs by Gas Chromatography, and ELISA Analysis

Feces were collected using a stool kit (Kalszyk, Poland) and then were stored at −80 °C until analysis. A 0.5 g fecal sample was suspended with 5 mL of water and mixed intensively for 5 min, then after pH adjustment to 2–3 (with a 5 M HCl solution), suspension was shaken and centrifuged (20 min at 5000 rpm). After that, the supernatant was filtered. Chromatographic analyses with a fused-silica capillary column (DB-FFAP, 30 m × 0.53 mm × 0.5 µm) were carried out using the Agilent Technologies 1260 A GC system (with a flame ionization detector (FID). The initial temperature was 100 °C (maintained for 0.5 min), then raised to 180 °C (ramping of 8 °C/min, maintained for 1 min, then to 200 °C (ramping 20 °C/min), and, finally, reached 200 °C and sustained for 5 min.

Fecal zonulin and calprotectin concentrations were determined by immunoenzymatic approach by means of commercially available kits (Immunodiagnostic tests; Germany).

### 2.5. Diet Analysis

Throughout the course of the study, diet samples were collected 20 times on randomly selected days to analyse meal composition. To obtain reference samples for industrial diets, 20 samples were collected from different batches. With the kitchen diet, 20 samples were collected from each meal separately: 20 breakfasts, 20 lunches, 20 dinners, and 20 suppers. Samples were collected in sealed containers (PROFILAB, Poland). The samples of kitchen and industrial diets were tested for protein, fat, and fiber content. The moisture, ash, protein, and fiber contents were determined according to the method described by AOAC [13,14]. Total insoluble and soluble dietary fiber was determined according to the enzymatic-gravimetric method using Fibertec 1023 (Tecator Tech., Sweden). Dry matter was determined by drying the samples in an oven at 105 °C until a constant weight was obtained. Ash was ascertained by incineration in a muffle furnace at 580 ºC for 8 h. Crude protein (N × 6.25) was measured by the Kjeldahl method, in a Büchi Distillation Unit B−324 (Büchi Labortechnik AG, Flawil, Switzerland). The Kjeldahl method was used, as described by AOAC (Helrich, 1990) [14]. Fat content was determined using the Soxhlet method, according to the methodology described in [15], and dietary fiber was measured enzymatically using a Megazyme kit, according to the K-TDFR 01/05 procedure (Megazyme International, Ireland) [16]. Samples were cooked with heat α-amylase, incubated at 60 °C with protease and amyloglucosidas, and treated with ethanol. The residue was filtered, dried, and weighed.

### 2.6. Statistical Analysis

Statistical analyses were carried out in StatView version 5.0 (SAS Institute Inc., Cary, NC, USA) and Statistica 13.1 (StatSoft, Kraków, Poland). The results were considered statistically significant at *p* < 0.05. Continuous variables distribution was tested for normality using the Shapiro–Wilk test. Quantitative data were presented as means ± SD, whereas for qualitative data, numbers and percentages were used. To assess the effects of the respective diets on the concentrations of zonulin, calprotectin, and prealbumin, the Kruskal–Wallis test was used. Additionally, to check whether other factors (the use of antibiotics, probiotics, or proton pump inhibitors) had an effect on zonulin, calprotectin, and prealbumin levels, the analysis of covariance (ANCOVA) was applied. For post hoc analyses the Conover test was adopted.

## 3. Results

### 3.1. Fiber and Protein Content

Fiber and protein contents in a kitchen diet declared by a manufacturer were higher compared to measures quantities. The amount of protein found in the samples accounted for 57% of the manufacturer’s estimation. With respect to fiber, the actual content represented 45% of the declared content. As for the industrial diets, no significant differences were found between measured and declared values (Table 2).

### 3.2. The Influence of Diet on Anthropometric, Biochemical and Gut Barrier Parameters

After 7 days of the kitchen diet implementation, we observed a reduction in body weight (by 1.0 kg), <0.05, and BMI (24.9 kg/m^2^ vs. 24.5 kg/m^2^) values, <0.05 (Table 3).

A decline in fecal zonulin was observed only in the kitchen diet group. In patients receiving the kitchen diet, butyric acid was slightly down and propionic acid went up. The most pronounced change was observed in patients on the industrial diet without fiber, whose levels of short-chain fatty acids declined most dramatically (Figure 2).

In the kitchen diet group, a rise in isocaproic acid was observed. In the group on the industrial diet with fiber, there was an increase in isovaleric acid, but also decreases in heptanoic, caproic, and valeric acids.

In the next stage of analyses, we evaluated whether the intake of drugs, commonly known to diminish gut barrier integrity parameters (proton pump inhibitors, antibiotics) might influence the results. We were able to show that calprotectin change was significantly affected by use of proton pump inhibitors (Table 4; Figure 3).

## 4. Discussion

In the diet composition analysis, we found differences between the declared fiber and protein contents compared to measured ones. Analogous differences were observed in the study by de Sousa et al. The authors found, e.g., that instead of the anticipated 22.76 g of protein, 100 mL of soup contained only 4.99 g of protein [17]. Similar observations were found by other authors [18,19,20]. The main reason for such discrepancies is the production method—the diet is first cooked, then blenderized and passed through a sieve. As the food is passed through a sieve, some meat fibers collect there and end up discarded. Additionally, the extended cooking time contributes to protein loss. According to Cupisti et al., when meat is cooked for 30 min, approx. 87.4% of the crude protein is retained [21]. In our study, differences were noted in the fiber content, too. This, again, is primarily due to the preparation method—cooking significantly reduces fiber content [21,22]. These findings are consistent with those made by Veira et al., where the kitchen diet provided smaller quantities of protein, fat, and energy [23]. According to Gallagher et al. [24], in order to maintain a stable body weight while the patient is fed with a kitchen diet, the energy content should be increased by about 50% [24]. Likewise, Santos et al. reported nutrient losses in homemade enteral diets by 50% [20]. When planning to introduce a homemade enteral diet, it is essential to address the substantial protein loss. Dietary protein intake needs to be given special attention in malnourished patients [25].

We found that in patients fed with the kitchen diet, intestinal barrier permeability was least affected (based on fecal zonulin determinations). This finding does not lend itself to discussion easily, as there are a small number of studies investigating changes in zonulin after a stroke and also in patients subjected to enteral nutrition. Lasek-Bal et al. report an increase in zonulin during a stroke. A significantly higher mean concentration of zonulin was observed in patients >65 years of age compared to younger patients and in patients with arterial hypertension compared to patients without the disease [26]. A post-stroke increase in zonulin has been demonstrated in an animal model [27]. It is not possible to explain conclusively why zonulin concentrations improved in the patients supported by the kitchen diet. Such an outcome may have been affected by the fiber content, which may be correlated with lower zonulin release [28]. Some other factors may be implicated, too: diet prior to the stroke [29] as well as concomitant hypertension [30], diabetes mellitus [31], or obesity [32].

Our study demonstrated that calprotectin levels were most significantly affected by the use of proton pump inhibitors. The adverse effect of proton pump inhibitors on increasing calprotectin secretion was also observed by Lundgren et al. [33] and Castellani et al. [34]. According to Lundgren et al., this may be associated with the inflammation-triggering effects of PPI use, driving up calprotectin, in itself a marker of inflammation. An even more likely explanation can be sought in the fact that PPI use induces bacterial overgrowth in the small intestine with a secondary response of the immune system in the gut. Bacterial overgrowth in the small intestine may result in a response of neutrophils to bacteria in the gut [33,34]. The adverse effects of proton pump inhibitors on gut microbiota have been confirmed in other studies [33,35]. Their effect may be two-pronged: First, via the effect of PPI medications on pH, which affects the diversity of gut microbiota. The second mechanism may be related to the impairment of the function of bacterial proton pumps [36].

In this study, changes in SCFAs were observed only in the patients fed with a kitchen diet—when re-examining the results, we observed changes in the content of butyric and propionic acids only among these stool probes. This finding confirms the positive effect of dietary fiber on the synthesis of SCFAs [37] but in contrast to our results, we did not observe a positive effect of fiber-enriched enteral nutrition on SCFAs synthesis like Schneider et al. [38].

After the industrial, fiber-enriched diet we noticed a depletion of selected branched fatty acids (BCFA): valeric, caproic, and heptanoic acids (but not isovaleric acid). The BCFA depletion in this group may indicate reduced protein fermentation in the intestine [39].

At the same time, in the kitchen diet group, the synthesis of isocaproic acid increased, which may suggest enhanced proteolysis in the large intestine (intestinal protein fermentation is known to stimulate the production of branched-chain fatty acids) [39].

To summarize the results of the barrier markers, it seems that the kitchen diet had the best effects on both zonulin release and SCFAs production (Figure 4). On the other hand, this group also presented with a significant decrease in butyric acid, which is one of the most important SCFAs. Still, as suggested by Whelan et al., a decrease in fecal butyrate may not be related to reduced synthesis, but rather an increased absorption and use by intestinal cells [39]. Potentially then, short-chain fatty acids could be utilised more efficiently by enterocytes and a smaller amount of SCFAs would end up in feces. This, in turn, may have resulted in improved tight junction function and the observed reduction in zonulin release.

The findings from our study, pointing out the positive effect of the kitchen diet on intestinal barrier function, appear to confirm the observations of other authors. Gregori et al. [40] noted a lower incidence of diarrhoea following a switch to a kitchen diet. Positive effects due to the transition from an industrial to a kitchen diet were also reported by Gallagher et al. [24].

In conclusion, it seems that a cooking diet based on cream soups can be a valuable addition to a stroke patient’s diet. It seems to be a valuable support for the gut microbiota. However, due to high protein losses, the diet should be based on an industrial diet supplemented with dietary fiber.

## 5. Conclusions

In patients who suffered an ischemic stroke, there is an increase in zonulin release, which is a sign of increased intestinal permeability, and in calprotectin levels, signifying enhanced intestinal inflammation in this patient group.

The kitchen diet had the best effect on intestinal permeability after an ischemic stroke but is characterized by a loss of protein content during production.

Treatment with proton pump inhibitors should only be used when necessary. This is because these agents increase calprotectin concentrations, indicating enhanced intestinal inflammation.

## Figures and Tables

**Figure 1 ijerph-19-06168-f001:**
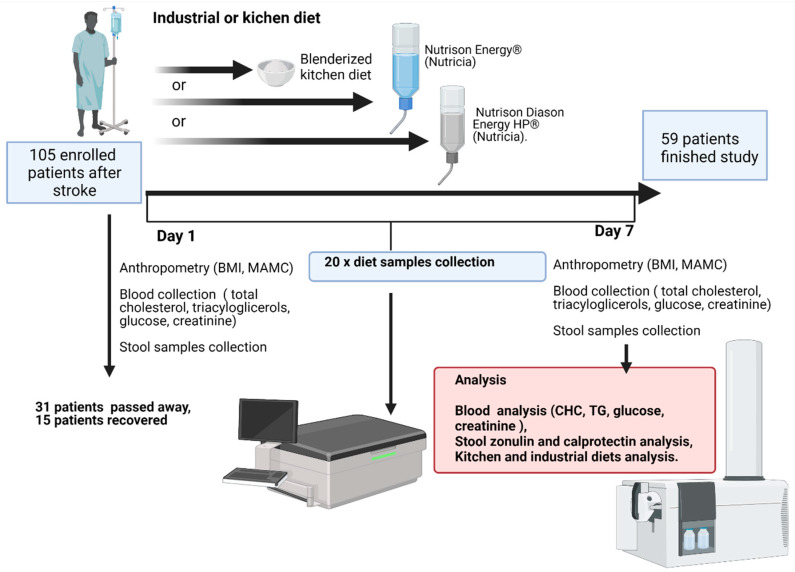
Study diagram. BMI—Body Mass Index, MAMC—mid-arm muscle circumference, CHC—cholesterol level, TG—triglycerides level.

**Figure 2 ijerph-19-06168-f002:**
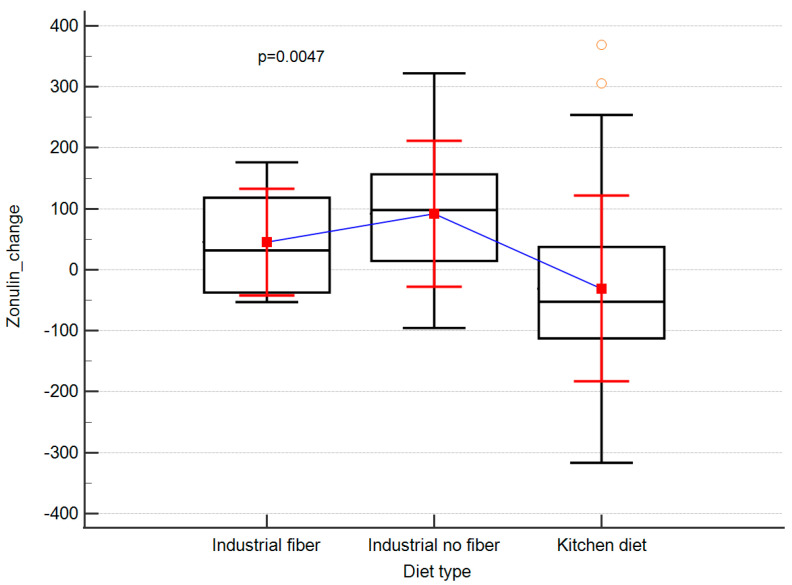
Effects of enteral diets on fecal zonulin release.

**Figure 3 ijerph-19-06168-f003:**
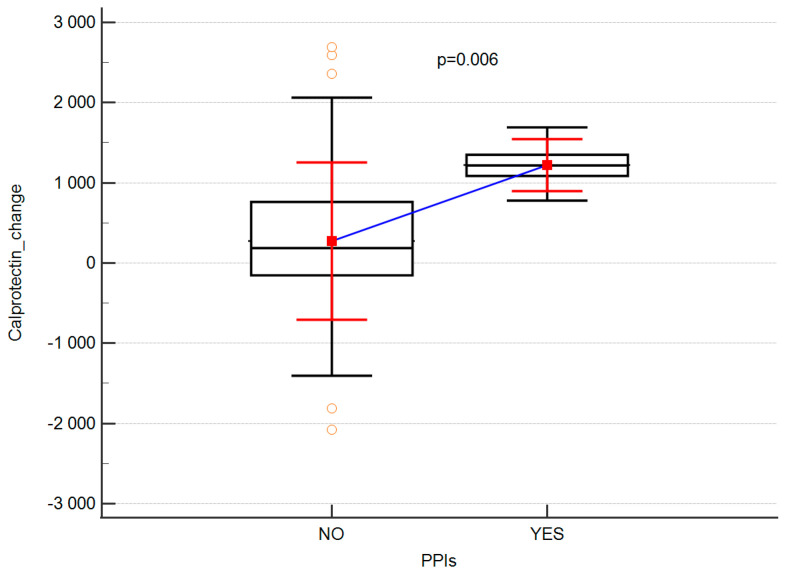
Effects of proton pump inhibitors on fecal calprotectin concentrations.

**Figure 4 ijerph-19-06168-f004:**
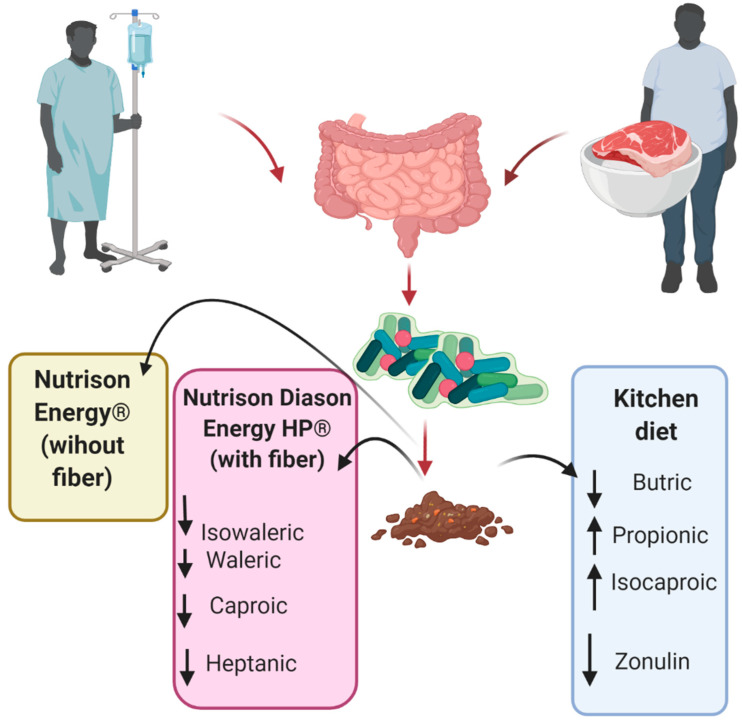
Overview of the results.

**Table 1 ijerph-19-06168-t001:** Comparison of blood biochemical parameters and anthropometric measurements in patients on Day 1 of the study.

	Blenderized Kitchen Diet	Nutrison Energy^®^	Nutrison Diason Energy^®^	*p*
BMI (kg/m^2^)	24.9 ± 24.9	24.6 ± 3.8	23.8 ± 3.8	ns
Body mass	64.7 ± 10.5	62.9 ± 12.0	63.1 ± 10.4	ns
MAMC	31.8 ± 4.5	29.7 ± 2.7	29.3 ± 2.9	ns
Calf circumference (cm)	34.4 ± 4.2	32.8 ± 3.7	33.4 ± 3.0	ns
Total cholesterol (mg/dL)	181 ± 49	160 ± 49	182 ± 53	ns
Triglycerides (mg/dL)	119 ± 63	130 ± 83	109 ± 37	ns
Glucose (mg/dL)	158 ± 49	131 ± 38	166 ± 43	ns
Creatinine (mg/dL)	0.9 ± 0.4	1.0 ± 0.3	1.0 ± 0.6	ns
Lymphocytes (tys/µL)	1.73 ± 0.78	1.5 ± 0.83	2.31 ± 1.5	ns
Erythrocytes (mln/µL)	4.67 ± 0.51	3.88 ± 0.88	4.96 ± 0.61	ns
Haemoglobin (g/dL)	13.95 ± 1.52	11.19 ± 2.51	14.18 ± 0.61	ns
CRP (mg/L)	31.23 ± 44.47	80.51 ± 68.05	58.27 ± 87.27	ns

MAMC—Mid-arm muscle circumference.

**Table 2 ijerph-19-06168-t002:** Nutritional content according to nutrition software vs. laboratory determinations in the diets.

	Dry Matter (%)	Protein (g)	Fat (g)	Total Fibre (g)	Ash
Kitchen diet, content determined in 100 mL	10.4%	2.4	4.5	1.1	0.68
Kitchen diet, content according to the software per 100 mL	No information	4.2	4.4	2.4	0.9
Nutrison Energy diet, content determined in 100mL	29%	5.4	5.2	0.02	0.92
Nutrison Energy, content declared by the manufacturer per 100 mL	No information	6.2	5.8	<0.1	No information on the packaging
Nutrison Diason Energy HP diet, content determined in 100 mL	28.10%	7.1	7.7	1.8	0.8
Nutrison Diason Energy HP diet, declared by the manufacturer per 100 mL	No information	7.7	7.7	1.5	No information on the packaging

**Table 3 ijerph-19-06168-t003:** Anthropometric and gut barrier parameters (short and branched fatty acids, zonulin, calprotectin) in studied groups by diet type and time of analysis (mean ± SD).

	Kitchen Diet before	Kitchen Diet after	Nutrison Diason Energy HP before	Nutrison Diason Energy HP after	Nutrison Energy before	Nutrison Energy after
	MEAN ± SD	MEAN ± SD	MEAN ± SD	MEAN ± SD	MEAN ± SD	MEAN ± SD
Lymphocytes [10^3^/µL]	1.73 ± 0.78 *	1.33 ± 0.66 *	2.31 ± 1.5	2.38 ± 1.8	1.5 ± 0.8	1.52 ± 0.48
Erythrocytes [10^6^/µL]	4.67 ± 0.51 *	4.27 ± 0.65 *	4.96 ± 0.61 *	4.3 ± 0.5 *	3.88 ± 0.9	4.96 ± 0.6
Haemoglobin [g/dL]	13.95 ± 1.5 *	12.8 ± 2.1 *	14.18 ± 1.5 *	12.3 ± 1.8 *	11.2 ± 10.6	11.22 ± 2.7
CRP [mg/L]	31.2 − 2.7 *	68.8 − 25.1 *	58.3 ± 11.7	46.2 ± 30.5	80.51 ± 78.3	77.04 ± 55.4
Body weight (kg)	64.7 ± 10.5 *	63.7 ± 9.9 *	63.1 ± 10.4	62.9 ± 10.5	62.9 ± 12.0	62.5 ± 12.1
BMI (kg/m^2^)	24.9 ± 4.2 *	24.5 ± 4.1 *	23.8 ± 3.8	23.1 ± 3.8	24.6 ± 3.8	23.7 ± 3.3
Calprotectin (ug/mL)	814 ± 629	1225 ± 162	1163 ± 830	1398 ± 805	1042 ± 1053	1376 ± 688
Zonulin (ng/mL)	410.3 ± 168 *	431.1 ± 137 *	421 ± 138	477 ± 161	397 ± 165	482 ± 138
Acetic acid (%)	27.9 ± 12.4	29.2 ± 7.6	35.9 ± 5.2	36.4 ± 4.7	31.5 ± 8.5	37.0 ± 11.9
Butyric acid (%)	20.2 * ± 7.2	16.1 * ± 5.9	16.8 ± 3.8	16.0 ± 2.7	14.6 ± 5.3	12.3 ± 6.6
Propionic acid (%)	16.3 * ± 5.8	18.9 * ± 6.2	17.9 ± 3.8	19.9 ± 7.6	18.4 ± 5.0	18.3 ± 7.1
Isobutyric acid (%)	7.9 ± 5.1	8.6 ± 7.6	5.7 ± 1.0	5.7 ± 1.2	6.6 ± 1.5	6.7 ± 2.0
Isovaleric acid (%)	15.0 ± 3.7	14.4 ± 3.3	12.4 * ± 2.3	12.8 * ± 29	13.8 ± 3.6	15.3 ± 4.4
Valeric acid (%)	7.7 ± 2.2	7.4 ± 2.1	7.2 * ± 0.8	6.7 * ± 1.4	7.3 ± 2.5	6.0 ± 2.2
Isocaproic acid (%)	1.2 * ± 27	1.7 * ± 1.5	1.5 ± 0.5	1.0 ± 0.4	2.6 ± 2.4	1.7 ± 2.2
Caproic acid (%)	2.5 ± 2.7	2.1 ± 1.7	1.6 * ± 0.7	0.8 * ± 0.3	2.7 ± 2.4	1.4 ± 2.0
Heptanoic acid (%)	1.3 ± 2.9	1.6 ± 1.8	1.1 * ± 0.6	0.7 * ± 0.4	2.53 ± 2.6	1.3 ± 2.2

BMI—Body Mass Index; SD—Standard Deviation. * *p* < 0.05.

**Table 4 ijerph-19-06168-t004:** Analysis of covariance ANCOVA for fecal calprotectin concentrations.

	SS	Degrees of Freedom	MS	F	*p*
Caloric intake	451,086	1	451,086	0.93634	0.338071
Type of diet	1,228,154	2	614,077	1.27467	0.288814
Antibiotics	127,430	1	127,430	0.26451	0.609397
Probiotics	49,523	1	49,523	0.1028	0.749889
PPI	2,819,281	1	2,819,281	5.85211	0.019398

PPI—proton pump inhibitors.

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
