# Peer review of "Kitchen Diet vs. Industrial Diets—Impact on Intestinal Barrier Parameters among Stroke Patients"

_ijerph, 2022, doi:10.3390/ijerph19106168_

Round 1

Reviewer 1 Report

In this article, the authors performed a comparison test between the use of kitchen diet and industrial diets with or without the addition of fiber in patients following ischemic stroke. The authors claim patients on the kitchen diet saw changes in the concentrations of lymphocytes, hemoglobin, erythrocytes, and serum concentrations of C-reactive protein but this is not supported in Table1. Is there data from Day 7 to support this claim?

This article is too short to be considered as such, please resubmit as a communication instead.

Introduction: The introduction is brief and to the point. However, please add information about zonulin and its function.

Methods: The statistical analysis section needs expanded. Please add the following:

1) Which post-hoc test was performed following Kruskal-Wallis, and which were the pairwise comparisons?

Results: Please fix the following issues:

1) Table 3: There is no table legend. Please explain the asterisk use. What are the percentages referenced to?

2) Figure 2: This figure has no y-axis label. The data representation can be improved via the use of box and whisker plots using Tukey’s representation, with outliers identified as dots. Please show the actual p-values for both the Kruskal-Wallis test and the post-hoc test.

3) Table 4: Please add the R-squared values as a measure of how well the model fits the data. If these values are low, please interpret the R-squared values with a scatter plot and change the analysis if needed.

4) Figure 3: Please add x- and y-axis labels.

Discussion: The discussion appears appropriate.

Conclusions: Please turn the conclusions into a paragraph of text rather than a numbered list.

Author Response

May, 11. 2022

Editors-in-Chief

IRERPH

Dear Editors,

I am taking the liberty to submit revised article to IERPH titled Kitchen diet vs industrial diets– impact on intestinal barrier parameters among stroke patients

Thank you for allowing us to resubmit an improved manuscript.The comments helped us to improve the quality of the manuscript. We considered all comments and recommendations and responded to Reviewers’ questions. The correction throughout the manuscript were done using red font.

Our responses to the reviews are attached below – marked in blue.

Thank you for your consideration. I look forward to hearing from you.

Sincerely,

Ewa Stachowska

Reviewer 1

In this article, the authors performed a comparison test between the use of kitchen diet and industrial diets with or without the addition of fiber in patients following ischemic stroke. The authors claim patients on the kitchen diet saw changes in the concentrations of lymphocytes, hemoglobin, erythrocytes, and serum concentrations of C-reactive protein but this is not supported in Table1. Is there data from Day 7 to support this claim?”

The point you are making is very valid. Yes, we have data from day 7. The data has been added to table 3.

“Introduction: The introduction is brief and to the point. However, please add information about zonulin and its function.”

Thank you. The information about zonulin and its function has been added.

“Which post-hoc test was performed following Kruskal-Wallis, and which were the pairwise comparisons?”

Thank you. The information has been added.

“ Table 3: There is no table legend. Please explain the asterisk use. What are the percentages referenced to?”

Thank you. The information has been added.

„Figure 2: This figure has no y-axis label. The data representation can be improved via the use of box and whisker plots using Tukey’s representation, with outliers identified as dots. Please show the actual p-values for both the Kruskal-Wallis test and the post-hoc test.”

Thank you. It has been amended

„Table 4: Please add the R-squared values as a measure of how well the model fits the data. If these values are low, please interpret the R-squared values with a scatter plot and change the analysis if needed.”

Thank you. It has been amended

„Figure 3: Please add x- and y-axis labels.”

Thank you. It has been amended

„Conclusions: Please turn the conclusions into a paragraph of text rather than a numbered list.”

Reviewer 2 Report

Major review:

1、It can be seen from the data in Table 3 that it should be represented by mean±SD. However, the present representation method should have a promising future and be further improved.

2、For the various short-chain fatty acids shown in Table 3, why the percentage change is used instead of the actual measured fatty acid content?We want to know what the ordinate is in Figure 2 and 3.

3、At present, according to the existing investigation,  it is difficult to show that different diets can regulate the intestinal mucosal barrier;

4、In Table 4 , no changes in the calprotectin release caused by the use of proton-pump inhibitors were found.The change in calprotectin concentrations is shown in Figure 3. The information of the graph should be further improved to make it more obvious.For example, we can further specify the number of people in each group and the concentration units of the vertical axis.

5.Please add to the discussion section the literature on why the use of proton-pump inhibitors may cause intestinal inflammatory responses.The literature on intestinal microbiota disturbance caused by the use of proton-pump inhibitors was added.line235-237

6、Update relevant literature information in time.just like Ref5、15、18、27.It is particularly important to maintain uniformity of document format.

Mini review:

  1. In the section of abstract, Language needs to be concise and use less for example(eg) or to reduce the explanatory content of indicators. See line 17-19.
  2. Line 30 decreased concentration of lymphocytes,....  as an inexact word of “concentration of lymphocytes”, it should be modified as “number of lymphocytes”
  3. Line55 short chain fatty acids (SCFA),shuold be as SCFAs; and the “acetate butyrate and propionate:shuold be added comma; a negative correlation
  4. Line60 “a negative correlation between ...”please confirm it ?negative or positive correlation;
  5. Line169  kitchen diet:. deleted:
  6. Line268 puréed soups .... what is the meaning of puréed?
  7. The English expression of this article needs further polishing.

Author Response

May, 11. 2022

Editors-in-Chief

IRERPH

Dear Editors,

I am taking the liberty to submit revised article to IERPH titled Kitchen diet vs industrial diets– impact on intestinal barrier parameters among stroke patients

Thank you for allowing us to resubmit an improved manuscript.The comments helped us to improve the quality of the manuscript. We considered all comments and recommendations and responded to Reviewers’ questions. The correction throughout the manuscript were done using red font.

Our responses to the reviews are attached below – marked in blue.

Thank you for your consideration. I look forward to hearing from you.

Sincerely,

Ewa Stachowska

„It can be seen from the data in Table 3 that it should be represented by mean±SD. However, the present representation method should have a promising future and be further improved.”

Thank you. It has been amended

„For the various short-chain fatty acids shown in Table 3, why the percentage change is used instead of the actual measured fatty acid content?”

Thank you for your question. That was the methodology. It is similarly described in „Skonieczna-Żydecka, Karolina et al. “Faecal Short Chain Fatty Acids Profile is Changed in Polish Depressive Women.” Nutrients vol. 10,12 1939. 7 Dec. 2018, doi:10.3390/nu10121939”

„At present, according to the existing investigation,  it is difficult to show that different diets can regulate the intestinal mucosal barrier;”

That is a valid comment.

In Table 4 , no changes in the calprotectin release caused by the use of proton-pump inhibitors were found. The change in calprotectin concentrations is shown in Figure 3. The information of the graph should be further improved to make it more obvious.For example, we can further specify the number of people in each group and the concentration units of the vertical axis.

Thank you. It has been amended

Please add to the discussion section the literature on why the use of proton-pump inhibitors may cause intestinal inflammatory responses.The literature on intestinal microbiota disturbance caused by the use of proton-pump inhibitors was added.line235-237

Thank you. It has been amended

Update relevant literature information in time.just like Ref5151827.It is particularly important to maintain uniformity of document format.

Thank you. It has been amended

In the section of abstract, Language needs to be concise and use less for example(eg) or to reduce the explanatory content of indicators. See line 17-19.

Thank you. It has been amended

Line 30 decreased concentration of lymphocytes,....  as an inexact word of “concentration of lymphocytes”, it should be modified as “number of lymphocytes”

Thank you. It has been amended

Line55 short chain fatty acids (SCFA),shuold be as SCFAs; and the “acetate butyrate and propionate:shuold be added comma; a negative correlation

Thank you. It has been amended

Line60 “a negative correlation between ...”please confirm it ?negative or positive correlation;

Thank you. It has been amended

Line169  kitchen diet:. deleted:

Thank you. It has been amended

Line268 puréed soups .... what is the meaning of puréed?

„Puréed soups” has been changed to „cream soups”. It is a soup with vegetable, meat, oil and cereals which is cooked and mixed. 

Round 2

Reviewer 1 Report

Figure 2 needs a y-axis label.

Author Response

Review Report

Reviwer 1

Thank you again for all your comments.

The description to Fig 2 has been completed

Reviewer 2 Report

I have  no question.

Author Response

Review Report

Reviwer 2

Thank you again for all your comments.